## RESEARCH ARTICLE

# *StrIPETrack*: a real-time, ROI-flexible tracking platform for high-throughput zebrafish behavior

**Camden E. Cummings, Brandon L. Bastien, Jaqueline A. Martinez, Jinyue Luo and Summer B. Thyme\***

### ABSTRACT

Quantitative phenotyping is essential to studies of animal behavior, enabling systematic analysis of variation arising from natural diversity or experimental manipulation. High-throughput behavioral assays that can simultaneously test multiple animals support sufficiently powered studies of behavioral variation, but accurate tracking of each animal is critical. Furthermore, behavioral tasks and experimental arenas span a wide range of complexity, from the reaction of a single larval zebrafish to an acoustic stimulus to associative conditioning in cue-rich environments. Here, we developed and validated *StrIPETrack* (Structural similarity-based Image Processing for Estimation and Tracking), a Python-based, modular animal tracking software designed for flexible region-of-interest definitions and extensibility across assays. We show that *StrIPETrack* measures activity comparably to our previous LabVIEW-based zebrafish tracking software and detects similar behavioral differences between wild-type clutches. In addition, *StrIPETrack* accurately captures behavior in a complex arena: the Y-maze. Our approach for analyzing Y-maze navigation yields an expanded set of metrics beyond turn count and direction, revealing more subtle behavioral variation. Overall, this versatile software can be applied to monitor the activity of multiple animals in parallel in both simple, high-throughput and more complex assays, and it can be readily adapted to new paradigms.

**KEY WORDS: Zebrafish, Y-maze, High-throughput behavior, Spatial navigation, Working memory, Open-source**

## INTRODUCTION

Animal behavior arises from the coordination of multiple physiological processes, including nervous, endocrine, immune, muscular, and cardiac systems. Accordingly, variation in behavior can provide a sensitive functional readout of the effects of genetic and environmental perturbations on an animal. In model organisms, characterizing how genetic or neuronal perturbations alter behavior can also provide insights into the neurobiological basis of neurological and neuropsychiatric conditions. Therefore, reproducible and robust behavioral assays, coupled with sensitive quantitative analysis methods, are critical for both basic and translational animal research. High-throughput behavioral phenotyping in small model organisms enables the screening of hundreds of genetic mutations or thousands of small molecules for effects on behavior (MacRae and Peterson, 2015; Marcogliese et al., 2022; Yemini et al., 2013).

Zebrafish, particularly at the larval stage, have emerged as a powerful vertebrate model for high-throughput behavioral phenotyping due to their small size, rapid development, and relatively conserved brain architecture (Mueller and Wullimann, 2009). Genetic screens of more than 100 mutants and pharmacological screens of over 10,000 compounds have been behaviorally characterized (Kokel et al., 2010; Thyme et al., 2019). Findings in zebrafish have also been translatable, as hits from zebrafish screens have entered clinical trials, such as for Dravet syndrome (Baraban et al., 2013; Patton et al., 2021). Previous scaling efforts have leveraged multi-well plates to measure sleep and circadian rhythms (Rihel et al., 2010), seizure-like behavior (Griffin et al., 2020), and responses to acoustic and visual stimuli (Randlett et al., 2019; Wolman et al., 2015).

Several software tools for real-time tracking of zebrafish exist. Bonsai (Lopes et al., 2015), an open-source visual programming language, and Zantiks, a proprietary tracking platform, both require adoption of software-specific programming environments. Our group's previously published zebrafish tracking software (Joo et al., 2020) was implemented in LabVIEW, which requires a paid license. Open-source and Python-based alternatives exist [e.g. Stytra (Štih et al., 2019)], but in practice are often optimized for a particular arena geometry, animal number, hardware configuration, or stimulus delivery. A modular, open-source platform paired with an extensible behavioral apparatus (Joo et al., 2020) provides the greatest flexibility for innovative behavioral assay development.

Scalable quantification of more complex behavioral tasks performed by juvenile zebrafish imposes software requirements that extend beyond those of commonly used larval assays. One such task is Y-maze navigation, which provides a measure of spatial working memory (Lalonde, 2002). One adaptation of this assay is the free-movement pattern (FMP) Y-maze (Cleal et al., 2021a,b, 2023). In this task, zebrafish freely explore an equilateral, three-armed Y-maze, and their pattern of turns is biased toward an alternation strategy (left-right-left-right/LRLR or right-left-right-left/RLRL) over other possible turn permutations. Pharmacological manipulation using drugs known to impair working memory disrupts performance of zebrafish in this assay (Cleal et al., 2021b). Notably, this bias is conserved across many vertebrate species, including humans (Cleal et al., 2021a).

To facilitate behavioral analysis across a wide array of zebrafish assays, we developed *StrIPETrack*, an open-source, Python-based tracking software. Our previously published zebrafish tracking system (Joo et al., 2020) was implemented in LabVIEW, which limits accessibility and is not easily extensible. *StrIPETrack* supports flexible region-of-interest (ROI) selection, enabling robust tracking of animals in simple and complex arenas. Position information can be processed downstream or in real time, supporting quantification of multiple metrics, such as spatial

Department of Biochemistry and Molecular Biotechnology, The University of Massachusetts Chan Medical School, Worcester, MA 01605, USA.

\*Author for correspondence (Summer.Thyme@umassmed.edu)

S.B.T., 0000-0003-3593-4148

preference within an arena and transitions between different zones. Here, we validate the tracking performance of *StrIPETrack* relative to our LabVIEW-based tracking in 96-well larval behavior assays and demonstrate its ability to accurately quantify turning behavior in the FMP Y-maze.

## RESULTS

### Development and capabilities of *StrIPETrack*

Python-based *StrIPETrack* functions by having the user select ROIs prior to the tracking process. Tracking is simplified by selecting the bounds of the areas where each fish will be. ROIs can be selected

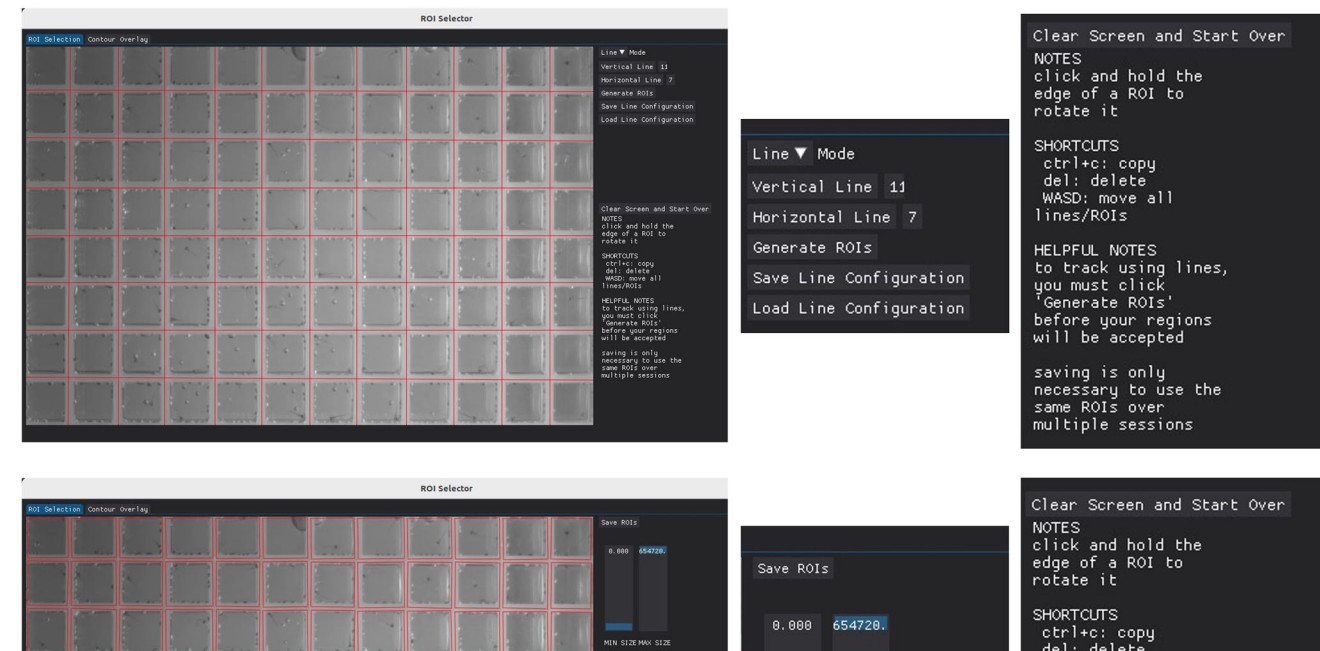

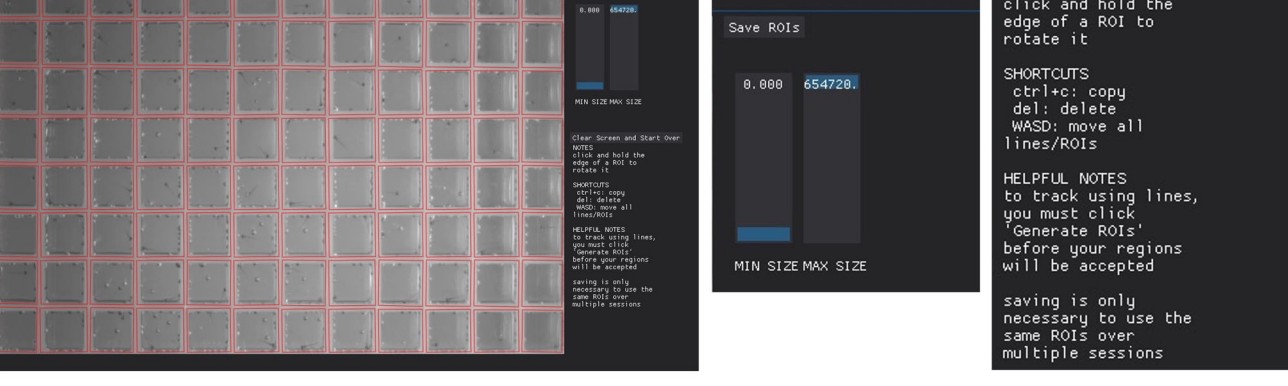

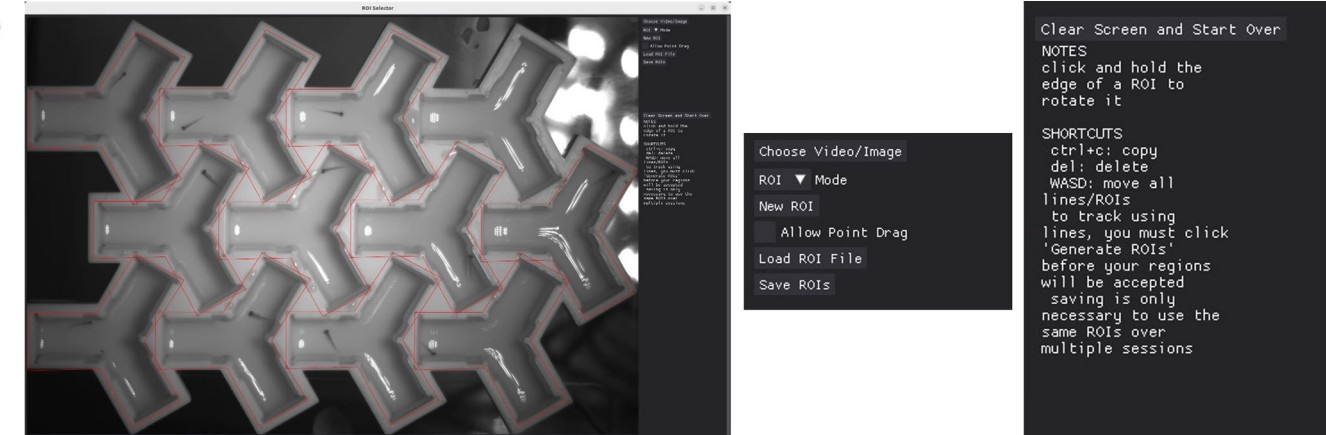

**Fig. 1. ROI preparation in *StrIPETrack*.** There are two ROI generation methods, and the same set of functions are allowed in both. For example, keyboard shortcuts move lines/polygons, delete, and copy. Line or polygon files can be saved and re-opened and modified later. For additional instruction, see the video in the GitHub repository. (A) The "Line" tool for convex ROIs. In "Line" mode, users can drag and drop lines or line edges. For a 96-well plate, a user would enter 11 into the 'Vertical Line' box, and 7 in the 'Horizontal Line' box. If any lines needed to be moved to match the cell walls, they could be shifted as a group with the WASD keys, or individually by with a drag and drop interface. Then, the user would hit 'Generate ROIs', making individual ROIs bound by each line. These ROIs can be moved, deleted, or changed as in the typical interface. (B) The "ROI" tool for concave ROIs. User selects each vertex with left click, and 'seals' ROI using right click. Once an ROI is complete, it can be moved using drag and drop interface, and rotated by selecting vertex and moving mouse.

manually, through a graphical user interface (GUI) (Fig. 1). In *StrIPETrack*, each operation (i.e. ROI selection, computer vision, Y-maze analysis) is an entirely separate module, not requiring the use of others. This design decision means the software is both easy to add to and easy to incorporate into an existing pipeline. As in LabVIEW, Python sends acoustic and visual stimuli commands to an Arduino and Teensy microcontroller (Joo et al., 2020).

For segmentation, we use structural similarity index measure (SSIM), an algorithm that assesses the similarity of two images (Wang et al., 2004). This assessment produces a difference image (i.e. a matrix of the detected difference; Fig. 2), an approach which has been widely used to identify animals for tracking (Chen et al., 2023; Loza et al., 2006). We apply SSIM to compare the current frame to a background frame, highlighting movement within each frame, as the difference between a background frame and the current frame is expected to correspond to the fish. To obtain the background image, we take a mode over the whole movie. If the fish is moving, it is excluded, resulting in an image that contains only the background. After we use this image to segment the fish and identify their likely positions, we apply a Kalman filter to smooth the position data. Additionally, we discard tracks that do not move over multiple frames. Through optimization of the SSIM implementation provided by the open-source library scikit-image (van der Walt et al., 2014) and alteration of the algorithm for real-time tracking applications, significant speed improvements were achieved (Table S1). Tracking at 30 frames per second was achieved on an Intel Core i7, 8GB RAM computer.

## Behavioral analysis software comparison

First, we compared behavioral metrics generated from our existing LabVIEW-based tracking and our custom Python software. We previously demonstrated that significant behavioral differences

between larval clutches from different parents can be detected (Joo et al., 2020). To benchmark *StrIPETrack* against LabVIEW, we placed two different clutches of wild-type larval fish in alternating columns of a 96-well plate and tracked their movement with both software. Two separate breeding pairs were crossed to generate two clutches of fish, and fish from these clutches were placed in alternate columns of our 96-well behavior plates. Over a 3-day time course with multiple stimuli (Capps et al., 2025; Joo et al., 2020), clear circadian-dependent activity differences were observed (Fig. 3A,B). Responses to dark flash were different between the two clutches. Specifically, frame-by-frame delta pixels (Δ pixels, dpix) in response to dark flashes were significantly higher in Clutch 1 than Clutch 2 in both methods (Fig. 3C,D). Thus, *StrIPETrack* captures similar behavioral variation comparable to LabVIEW, and is sufficiently sensitive to detect differences between groups of larval zebrafish.

We next evaluated the extent to which *StrIPETrack* and other available tracking platforms produced comparable animal-tracking outputs. To compare *StrIPETrack* directly with LabVIEW, we used *StrIPETrack* to analyze a 30-min video generated by LabVIEW and then assessed the correlation of the dpix measured by the two programs. There was a strong, positive correlation between the dpix calculated by *StrIPETrack* and LabVIEW tracking (r=0.973, *P*<0.0001) (Fig. S1). Consistency was moderate between the two tracking software [ICC(3,1)=0.513, *P*<0.0001]. We also compare the tracking capability of *StrIPETrack* against both BonZeb (Guilbeault et al., 2021), a zebrafish-specific plugin for Bonsai (Lopes et al., 2015), and Stytra (Štih et al., 2019). Although both BonZeb and Stytra are highly customizable and useful for multi-animal tracking, neither performed well with the Y-maze dataset described below (Table S2). BonZeb performed better than Stytra

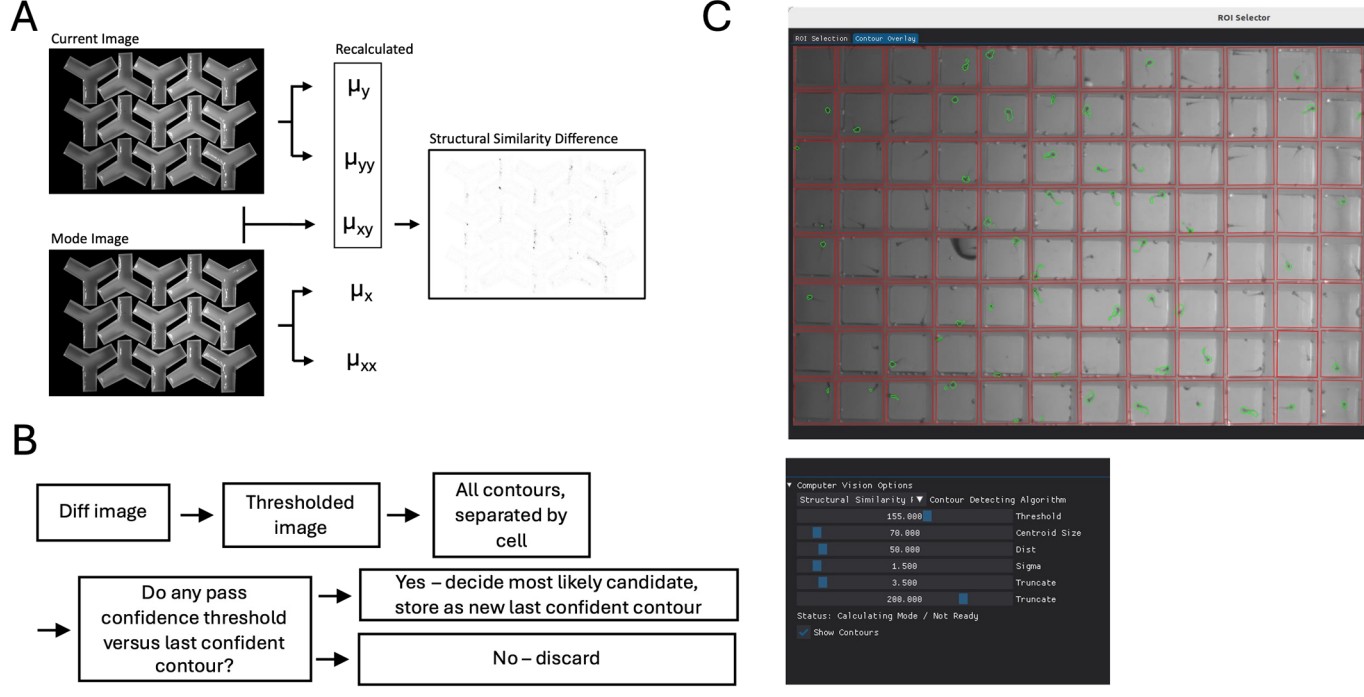

**Fig. 2. Structural similarity implementation.** (A) In structural similarity, matrices $\mu_y$ and $\mu_{yy}$ are calculated based on image 1, matrices $\mu_x$ and $\mu_{xx}$ are calculated based on image 2, and matrix $\mu_{xy}$ is calculated from both images. Because the sci-kit implementation of the algorithm does not anticipate a relationship between the two images, these values are recalculated each time. When the prior image is used, the current image's $\mu_x$ and $\mu_{xx}$ will become $\mu_y$ and $\mu_{yy}$. When mode is image 1, $\mu_x$ and $\mu_{xx}$ can be stored. (B) The difference image given by SSIM is thresholded, and contours for each cell are found. Each is checked for confidence, and if passes, is stored as new contour. (C) Detected zebrafish movements are highlighted in green, based on selection of the "Show Contours" checkbox in the GUI.

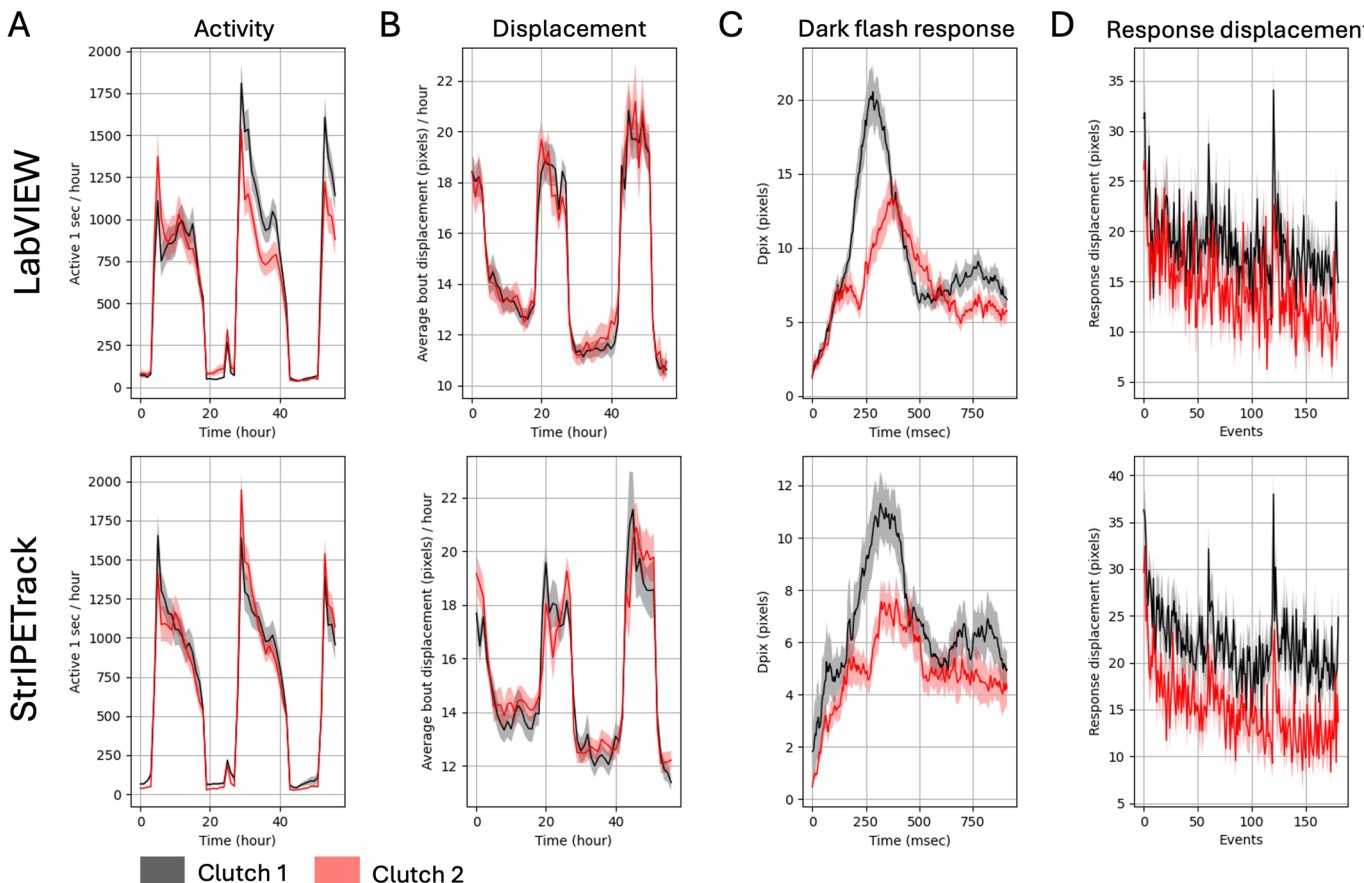

**Fig. 3. Comparison of 96-well plate larval behavior tracking between LabVIEW and *StrIPETrack*.** (A) Active seconds per hour, a measure of overall activity. (B) Average bout displacement. This measure requires accurate tracking of the X-Y position. (C) Response trace for all dark flashes. Three blocks of 60 1-second-long dark flashes were delivered once per minute, with an hour break in between the blocks. Dark flash tracking is performed post-hoc and remains consistent between *StrIPETrack* and LabVIEW; numerical differences are likely explained by lighting differences between behavior tracking boxes. Kruskal–Wallis ANOVA *P*-value LabVIEW=0.0006, *StrIPETrack*=0.001. (D) Response displacement for each dark flash. Kruskal–Wallis ANOVA *P*-value LabVIEW=2.2e-5, *StrIPETrack*=4.8e-8. LabVIEW clutch 1 *N*=45, clutch 2 *N*=43; *StrIPETrack* clutch 1 *N*=41, clutch 2 *N*=48.

but tracked many non-existent turns because the fish position was inaccurately switching arms.

### Free-movement pattern (FMP) Y-maze analysis adaptability
To test the compatibility of *StrIPETrack* with more complex ROIs, we used 3D-printed Y-mazes to adapt our behavioral apparatus to the previously published FMP Y-maze task (Fig. 2A, Fig. S2). The common behavioral metric quantified in FMP Y-mazes is the turn direction into other arms (either left or right from the current arm) (Cleal et al., 2023). Sequences of four consecutive turns, called tetragrams, are calculated, and a preference for alternation is conserved across vertebrate species (Cleal et al., 2021a). To evaluate the accuracy of our code, we compared turns calculated by our software to turns measured by hand in a subset of fish at 10-, 15-, 21-, and 27-days post-fertilization (dpf) using a Pearson correlation. There was a strong, significant positive correlation between the two methods (r=0.903, *P*<0.0001), and inter-method agreement was high [ICC(2,1)=0.899, *P*<0.0001] (Fig. 4A). We found no systematic difference between our software and manual counting (Fig. 4B). To validate the robustness of *StrIPETrack*, we additionally analyzed a recording with lower contrast and found that the zebrafish were successfully tracked (Fig. S3) (r=0.807, *P*=0.0014). Thus, our code can accurately quantify tetragrams, highlighting its adaptability to complex ROIs.

We next compared the performance of zebrafish in the FMP Y-maze across ages. When analyzing alternation tetragram percentages, all ages performed alternation tetragrams above random chance (Fig. 5A). To determine if performance in the task varied across time, we binned tetragrams performed into 10-min bins. Zebrafish starting at 17 dpf performed alternating tetragrams in each bin above random chance. In contrast, the 10 dpf fish did trend toward alternations above random chance but were unable to reach false discovery rate-corrected in any of the 10-min bins (Fig. 5B,C). To assess for variability across wild-type fish, we compared alternation tetragram percentages across wild-type siblings from four different lines and found the LRLR/RLRL peak percentages ranged from as high as about 60% to as low as about 20% in 21 dpf fish. These percentages were always above random chance (Fig. S4). We asked whether a simple biometric could predict performance in the task. We grouped the top 50% and lowest 50% of fish of 21 dpf fish based on body size and tested whether either group performed better or worse than the other. The top 50% of the fish performed the task better than the lowest 50% of fish (main effect of size: *P*=0.000011), indicating that size could be a predictor of working memory performance (Fig. S5). We also stratified the 10 dpf fish into the upper and lower 50% of fish based on body size, and, like the 21 dpf fish, the bigger 10 dpf performed the task better than the smaller fish (main effect of size: *P*=0.011594).

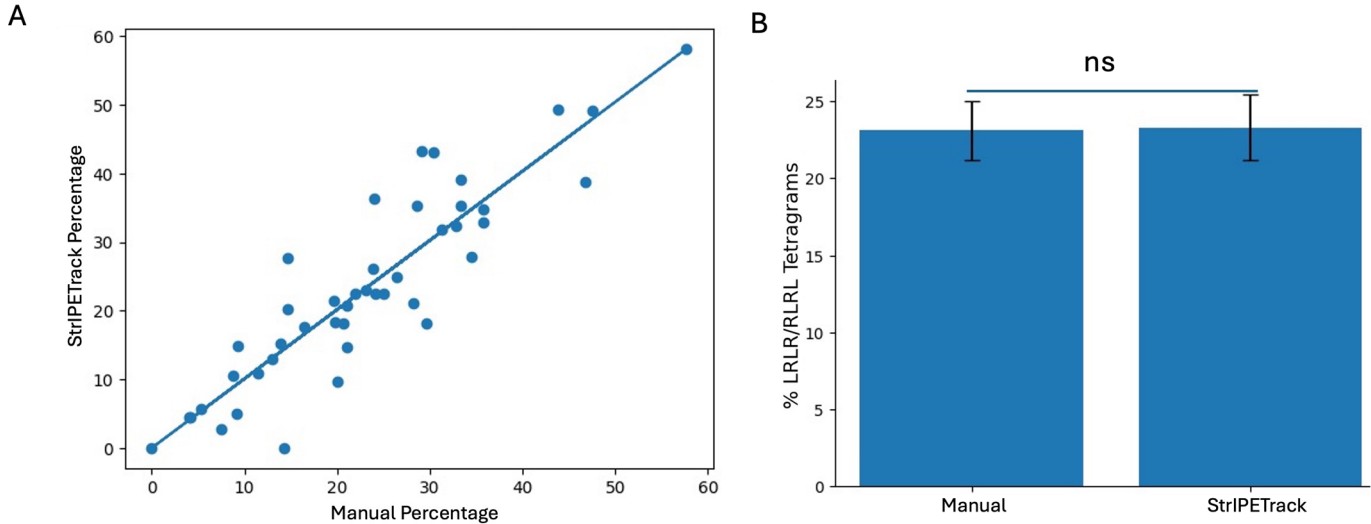

**Fig. 4. Validation of tetragram accuracy of Python tracking software.** (A) Scatterplot plotting alternation tetragram (LRLR/RLRL) percent of individual fish measured by manual counting and our custom Python tracking software. Line represents the line of best fit [Pearson correlation: $r=0.903$, $r^2= 0.82$, $P<0.0001$; ICC(2,1) 0.899, $P<0.0001$, $n=44$]. (B) Bar graph comparing the alternation tetragram percentages for individual fish measured by manual counting and our custom Python tracking software. ns, not significant result of paired $t$-test: $t=-0.22$, $P=0.82$, $n=44$.

Lastly, because our code was successfully able to quantify other behavioral metrics in larval behavior, we wanted to generate other behavioral metrics not previously quantified in Y-maze behavioral analysis pipelines. In addition to total tetragram numbers, which have been used in previous analyses, we quantified velocity, arm preference, and generated heatmaps representing position in the Y-maze (Fig. 5D,E). Total tetragrams and velocity provide readouts of overall activity, and the arm preference and heat maps provide readouts of arm bias, such as preference for the arm that the fish habituated in. These readouts supplement the tetragram percentage analysis by providing additional behavioral profiles and enhancing the interpretation of Y-maze behavior.

## DISCUSSION

Here, we developed and validated an open-source, Python-based animal tracking software that is adaptable to simple and complex behavioral assays. Tracking larval zebrafish with *StrIPETrack* produced behavioral metrics comparable to those obtained with our established LabVIEW-based program (Joo et al., 2020), including the detection of significant differences between wild-type clutches (Fig. 3D). The magnitude of these differences reinforces the necessity of using sibling controls to achieve accurate results in zebrafish experiments. Leveraging the software's ROI flexibility, we increased the capacity of the Y-maze assay to 12 arenas in a single plate that can be recorded simultaneously while being tracked independently. With multiple custom-built behavioral arenas, more than 100 animals can be assessed in a single day. This scalability provides substantial potential for screen-based studies, such as investigating hundreds of mutant lines associated with neuropsychiatric disorders (Thyme et al., 2019) or screening compound libraries (MacRae and Peterson, 2015). Thus, this modular software supports standard larval behavior in 96-well plates and the generation of rich data sets for more complex paradigms.

Compared to other software, *StrIPETrack* was able to accurately track zebrafish in a variety of behavioral paradigms. Similar to our previous LabVIEW-based program (Joo et al., 2020), it also detected subtle differences in larval behavior. *StrIPETrack* was also able to quantify tetragrams in a complex juvenile navigation task, while neither BonZeb nor Stytra achieved comparable accuracy (Table S2).

Because both Bonsai and Stytra are highly customizable, it is possible that a different set of parameters or a modification of the code used would improve the results; however, the default settings did not yield robust tracking. Other options are available, such as machine learning-based approaches (Barreiros et al., 2021; Mathis et al., 2018; Teicher et al., 2025), although they also have constraints. Machine learning typically requires substantial training and the creation of annotated datasets prior to tracking, and these methods can be too resource-intensive for real-time tracking. In contrast, *StrIPETrack* performs well with limited computational resources and does not need to be trained. For quantifying more complex behaviors that cannot be fully captured by simple object tracking, such as posture dynamics, deep learning and semantic segmentation approaches may be more suitable.

Using our updated FMP Y-maze analysis, we expanded both the range of extracted behavioral metrics and our understanding of factors influencing task performance. Previous analyses of the FMP Y-maze mainly focused on alternation strategy by measuring the number of turns and percentage of tetragrams, which may overlook more subtle behaviors. In addition to these turn-based metrics, we quantify features such as velocity, zone preference, and activity (Fig. 5D). Our updated design includes a comb (Fig. S2), which synchronized the timing of fish navigation following habituation in one arm of the arena. Performance in the task was affected by age, with 10 dpf fish failing to perform tetragrams above random chance (Fig. 5C), consistent with previous reports (Cleal et al., 2023). Body size also affected performance at 21 dpf (Fig. S5), with larger fish outperforming smaller individuals. Although causality is difficult to infer, improved working memory in larger fish could enhance hunting, allowing them to outcompete siblings. Arena size represents another variable that likely influences performance and should be scaled accordingly with age, particularly when testing fish older than 24 or younger than 10 dpf.

Beyond the assays tested here, *StrIPETrack* can facilitate other complex behavioral paradigms. For example, in social preference assays, the social stimulus fish is not always monitored (Capps et al., 2025; Dreosti et al., 2015; Geng et al., 2022), and our flexible ROI selection can easily track both fish. A particular strength of

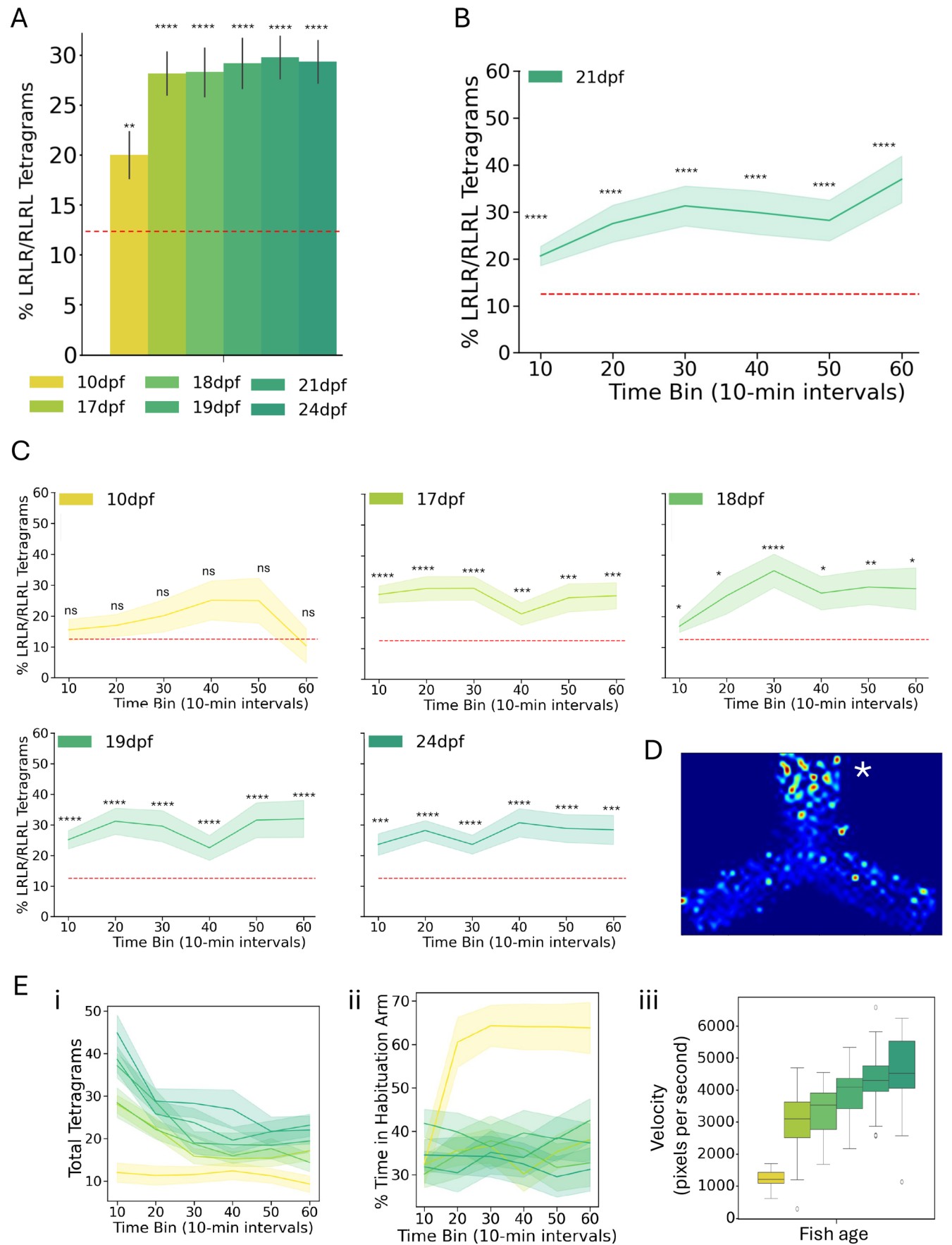

**Fig. 5.** See next page for legend.

**Fig. 5. Behavioral metrics of wild-type fish of different ages.** (A) Bar plot showing total alternation (LRLR/RLRL) tetragrams of wild-type zebrafish at different ages. (B) Exemplar graph showing alternation tetragrams binned in 10-min bins across the 1-h recording. (C) Line graphs showing other ages alternation tetragrams in 10-min bins. Statistical comparisons were made through one-sample $t$-tests against random chance for eight possible turn permutations (12.5%) and corrected for multiple comparisons using FDR. ns, not significant, *<0.05, **<0.01, ***<0.001, ****<0.0001. (D) Heat map of localization of one fish in Y-maze. (E) Other behavioral metric, including (i) total tetragrams, (ii) time spent in habituation arm, (iii) velocity. Error bars represent s.e.m. in A-C and Ei and Eii, and IQR in Eiii. Sample sizes of A, B, C, and E are as follows (10 dpf $n$=10, 17 dpf $n$=60, 18 dpf $n$=24, 19 dpf $n$=48, 21 dpf $n$=36, 24 dpf $n$=36).

this tracking software is also the ability to track fish in real time. By tracking several fish in parallel in real-time, assays where fish automatically receive stimulation upon entering a region of an arena (e.g. a specific arm of a Y-maze) become possible using this software. This capability can allow for higher throughput testing of learning and memory-based tasks, such as aversive conditioning (Aoki et al., 2015). Additionally, it is easily extensible to most aquatic species, and its modularity would support other real-time or 3D tracking tasks (e.g. novel tank assay) with minimal modification.

Taken together, we have developed and validated *StrIPETrack*, a modular tracking software that provides expanded behavioral metrics in simple and complex tasks across a wide range of fish ages. Robust and reproducible behavior tracking is essential for identifying the underlying sources of variation in animal behavior. This software is highly adaptable to complex arenas and behavior analysis pipelines, representing a valuable resource for animal behavior studies.

## MATERIALS AND METHODS
### Zebrafish husbandry
Zebrafish experiments were approved by the UMass Chan Institutional Animal Care and Use Committee (IACUC protocol 202300000053). All zebrafish were of the Ekkwill (EK) background. Animals were maintained on a 14 h/10 h light/dark cycle at 28°C. Larval fish were grown in 150 mm Petri dishes until 5 dpf, or until used for larval behavior assays, at a density of less than 160 fish per dish. Debris was removed from the dish prior to 4 dpf. Only larvae that were healthy with developed swim bladders were used in this study. After 5 dpf, fish were maintained in 8700 ml tanks at a density lower than 50 fish/tank, unless otherwise noted. Larvae at 5 dpf were fed live rotifers until 12 dpf. At 12 dpf, under slow water drip, hatched artemia and GEMMA 75 were fed twice per day until 60 dpf, and once per day on weekends and holidays.

### Larval behavior assays
Larval behavior assays were conducted as previously described (Capps et al., 2025; Joo et al., 2020). The custom-built behavior system consists of an infrared light source and camera overhead an acrylic platform. Larval zebrafish aged 4 dpf were placed in individual wells of a 96-well plate and sealed as previously described using a clear optical adhesive film (Thermo Fisher Scientific, 4311971). Plates were placed in the box in a 3D-printed fish plate holder. Our larval behavior pipeline includes dark flashes, light flashes, acoustic prepulse inhibition, and acoustic habituation. Acoustic and visual stimuli are controlled through a printed circuit board (PCB). A Grasshopper GigE camera with 50 mm fixed focal length lens with an IR filter was used for tracking, capturing fish illuminated with a long-range IR (850 nm) light. The light level from the white LED panel can be controlled as needed (Joo et al., 2020) (e.g. to induce a dark flash), and the baseline illumination level was approximately 410 lux in the larval behavioral assays. As previously described (Cleal et al., 2021a), the Y-maze assays were run in the dark, which was shown to have minimal effect on navigation and yield consistent results. Each experiment followed a file that dictated the times and durations of stimuli. We include a version of the PCB schematic, which has been updated for Teensy 4.1, as the Teensy 3.6 used in the original paper (Joo et al., 2020) is no longer available.

### Y-maze assays
To fabricate our Y-mazes, we use the 3D-printed models shown in Fig. 1B, which are available on GitHub. We use the comb shown in Fig. S1 to block the arms, so the fish habituate in one arm for 10 min prior to the beginning of the study. We printed the Y-mazes using white ABS (acrylonitrile butadiene styrene), as it is both resilient and cost-effective. To avoid creating visual landmarks for the fish, the indentation for the comb is copied to each arm as well. We provide three versions of the comb: one for each arm. The indentation shown at the end of each Y-maze's arm can be used to provide a stimulus for the fish, such as a colored piece of plastic.

Y-maze assays were performed in the same behavior boxes as the larval behavior assays. A single zebrafish was placed in the individual arms of a Y-maze that was blocked off with a divider for 10 min. Afterwards, the divider was lifted, and the fish was allowed to freely explore the Y-maze for 1 h. Videos were recorded using the FLIR FlyCapture SDK36 software at 30 frames per second with H.264 compression using a streaming capture setting.

### Data analysis
#### LabVIEW analysis of larval behavior
We used the code provided in Joo et al. (2020) as a comparison, as it demonstrates the expected behavior of *StrIPETrack*. It is a LabVIEW-based code for tracking, using the Visual Development Toolkit. It sends out stimuli at the appropriate times and tracks the delta pixels moved and the current position of each fish.

#### Manual tracking of Y-maze tetragrams
To validate the accuracy of the software's ability to detect entries into arms of the Y-mazes, we took a selection of random videos, spread across all ages of our data set, and manually tracked when fish entered and exited each arm. The data were analyzed to generate tetragrams, which were compared to the tetragrams generated by the tracking software.

### Statistics
Statistical tests were conducted in Python. To compare *StrIPETrack*'s ability to calculate tetragrams, *StrIPETrack*-generated tetragrams and manually counted tetragrams were compared using a Pearson's correlation, and agreement was assessed two-way random-effects intraclass correlation coefficient for absolute agreement (ICC2). To assess systemic bias between manual counting and *StrIPETrack*, a paired $t$-test was performed. Comparisons of alternation percent tetragrams to random chance (12.5%) were assessed using a one-sample $t$-test, corrected for multiple comparisons using FDR adjustments. A two-way ANOVA was used to compare the top 50% and bottom 50% of zebrafish by body size and, with a significant main effect of size, pairwise comparisons were calculated at each time point. Packages used for analysis include pingouin, seaborn, statsmodels.stats.multitest, scipy.stats, statsmodels.api and statsmodels.formula.api.

### Acknowledgements
We thank the UMass Chan fish facility staff and the Research Computing team, as well as members of the Thyme lab and Phil Campbell for helpful feedback on the software.

### Competing interests
The authors declare no competing or financial interests.

### Author contributions
Conceptualization: C.E.C., J.L., S.B.T.; Data curation: C.E.C., B.L.B.; Formal analysis: C.E.C., B.L.B.; Funding acquisition: S.B.T.; Investigation: C.E.C., B.L.B., J.A.M., J.L.; Methodology: C.E.C., B.L.B.; Project administration: S.B.T.; Resources: S.B.T.; Software: C.E.C., B.L.B.; Supervision: S.B.T.; Validation: C.E.C., B.L.B.; Visualization: C.E.C., B.L.B.; Writing – original draft: C.E.C., B.L.B., S.B.T.; Writing – review & editing: C.E.C., B.L.B., J.A.M., J.L., S.B.T.

### Funding
This research was funded by the following sources: Simons Foundation SFARI Pilot Award (SBT). National Institutes of Health [HD115159 and DP2NS132107 to S.B.T.]. Open Access funding provided by UMass Chan Medical School. Deposited in PMC for immediate release.

## Data and resource availability

All data are available in the main text, the supplementary information, or appropriate databases. Code is available from https://github.com/camden-cummings/stripetrack and https://github.com/camden-cummings/behavior-analysis-scripts. Files for generating 3D-printed arenas and updated PCB are available from https://github.com/camden-cummings/stripetrack_design_files. Implementation details of our computer vision algorithm are available from https://github.com/camden-cummings/strsim_for_speed. Our ROI selection GUI and associated instructional video are available from https://github.com/camden-cummings/roi_selector_dearpygui. Examples of analysis of Y-maze data are available from https://github.com/camden-cummings/y-maze-visualization.

## Peer review history

The peer review history is available online at https://journals.biologists.com/bio/lookup/doi/10.1242/bio.062503.reviewer-comments.pdf

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
