## [Peer Review File · Biology Open]

StrIPETrack: a real-time, ROI-flexible tracking platform for high-throughput zebrafish behavior

Camden E. Cummings, Brandon L. Bastien, Jaqueline A. Martinez, Jinyue Luo and Summer B. Thyme

10.1242/bio.062503

Editor: Valentina di Santo

Review timeline

Original submission: 29 January 2026

Editorial decision: 3 February 2026

First revision received: 28 March 2026

Accepted: 30 March 2026

Original submission

First decision letter

MS ID#: bio.062503

MS Title: StrIPETrack: a real-time, ROI-flexible tracking platform for high-throughput zebrafish behavior

Authors: Camden E. Cummings, Brandon L. Bastien, Jaqueline A. Martinez, Jinyue Luo and Summer B. Thyme

I have now reached a decision on the above manuscript.

The reviewer reports are shown at the bottom of this email.

As you will see, the reviewers gave favourable reports, but raised some critical points that will require amendments to your manuscript. I hope that you will be able to carry these out, because we would like to be able to accept your paper.

At this stage, we also ask you to ensure your manuscript complies with our formatting guidelines - please see our manuscript preparation guidelines for details. Provided you are able to fully address the referees' comments, we are positive about publication of your paper (we accept over 95% of revision submissions) and therefore hope you won't mind any extra work involved in reformatting your manuscript at this point.

Please upload both a 'clean' version of your Word file, along with a highlighted version clearly showing where you have made changes in the revised manuscript. Please avoid using 'Track changes' in Word files as these are lost in PDF conversion.

I should be grateful if you would also provide a point-by-point response detailing how you have dealt with the points raised by the reviewers in the 'Response to Reviewers' box. Please attend to all of the reviewers' comments. If you do not agree with any of their criticisms or suggestions please explain clearly why this is so.

Reviewer 1

Comments for the author

Overall, this is a well written paper describing a new open-source software tool for high throughput tracking of zebrafish. The tool developed looks to be an improvement over prior work from the lab that relied on the use of proprietary software and thus promises to be more widely accessible. The authors use the tool to analyze locomotor activity over an extended period of time, in response to 'dark flashes', and during the Y-maze. The data are largely clear and compelling that the tool is successfully working.

Major points

It would strengthen the paper quite a bit to include additional analysis and discussion of the comparison between StrIPETrack and the prior system developed by this group using LabView (Joo et al, 2020). While it's clear that both systems yield similar findings with respect to analyzing behavior in two clutches (Fig. 3), it is also clear from the figures that there are some differences in the shapes and variation in the data. For example, with the dark flash response, the numerical values are much higher using the LabVIEW approach versus StrIPETrack, and the variation seems larger with StrIPETrack. Other subtle differences, for example in the activity on the second bout also look different. While I do not think this is an issue for use of StrIPETrack, it behooves additional analysis such as: how correlated are the findings from the two systems on a more granular basis (e.g., at the second by second timescale), and why are the absolute values so different for the dark flash response? I think this is an opportunity to discuss what is similar and different in terms of the algorithms used for tracking that may lead to these differences. This is important because it is rarely discussed in the literature how specific algorithms may bias the data in one direction or the other. Using the data the author's have generated would make the paper more impactful by providing a direct comparison and discussion of different approaches to tracking.

The author's find that behavior can differ subtly from clutch to clutch. I think this is an important finding (that replicates their previous work) because it strongly suggests intra-clutch controls are essential. However, what is less clear to me is how much of the difference in behavior is driven by factors within the experimenters control. For example, in figure S3 the author's show that body size influences FMP Y-maze performance. I also see that the conditions for raising the fish are not well standardized (e.g, it's stated that fish are put into 150 mm dishes at a density of less than 160 fish per dish and then lower than 50/tank at 5dpf). Is it possible that fish in less dense dishes grew at a faster rate than those raised at higher density? Could this account or contribute to the clutch-to-clutch variability? I think this point needs to be discussed to better interpret this data.

A better measure to capture the consistency between manual and StrIPETrack in Fig. 4 would be interrater reliability via the intraclass correlation coefficient (ICC). I suggest including this in addition to, or in lieu of, the Pearson's R.

The ReadMe file at <https://github.com/camden-cummings/behavior-analysis-scripts> leaves a bit to be desired and should be expanded a bit to make it easier for users to use that do not have a lot of Python experience to use.

While not necessary, I would recommend including a YouTube tutorial or something similar to post on the github page to help potentially new users familiarize themselves with this tool. Otherwise, the sparseness of the github page will likely be a barrier for those not as familiar with computers to engage with the tool.

Minor points

How tedious is it for the user to draw all the different ROI's for either a 96 well plate or the Y-maze? It wasn't clear to me how much upfront time for the user would be needed to achieve what the authors have done.

How old were the fish used in Fig. 3? I did not see this in the methods

For the fish in Fig. 3, are these fish from different clutches from the same breeders? Or completely different breeders? If from different breeders, were they obtained on the same day and raised in parallel? More detail here would help better interpret these experiments.

No strain or source of the fish is indicated in the methods.

What are the ages and strains of fish in Fig. S2

What are the n's for Fig. S2, and what do the error ribbons represent?

What type of camera was used for capturing the videos.

Line 27 of pg. 9, what does ABS stand for? Please spell it out if it is not going to be used anywhere else in the paper (I didn't see it anywhere else).

In the caption for Fig. 5 it says error bars represent SEM. But what about the error ribbons for parts B through E, and in E iii this looks like box and whisker plots which is usually something like median and interquartile range; the edge of the boxes and whiskers should be clearly defined.

In Figure 5, it's stated that one-sample t-tests were used to compare against random (12.5%), were these corrected for multiple comparisons? The text of the paper seems to suggest this (pg. 6, lines 3-5), but the figure caption does not.

Given the spread in the use of open-source machine learning approaches to tracking (e.g., DeepLabCut) in the past 7-8 years, a discussion in why this system is different than such popular systems would be useful for readers as they make a decision about what approach to take.

Reviewer 2

Comments for the author

This manuscript presents StrIPETrack, a Python based, modular animal tracking software designed for flexible use in both simple and complex behavioral assays. The authors validate the software by comparing it against their previous LabVIEW-based pipeline in larval zebrafish assays and demonstrate its applicability to Y maze behavioral tasks. The topic is timely, the manuscript is very well written, and the software represents a valuable tool for behavioral biologists.

I review this work as a regular end user of animal tracking systems rather than a software development specialist (!), so my comments focus primarily on usability, clarity, methodological transparency, and how convincingly the validation demonstrates practical reliability.

Overall, in my opinion, this is a very strong contribution. My comments are mostly minor, with a few main suggestions for strengthening clarity and scope:

1. While the biological findings (age related differences, size effects, etc.) are interesting, I feel that discussing these distracts from the primary purpose—validating the tracking software. I would find it more useful to expand the discussion of potential software performance in other species (fish vs. non fish, aquatic vs. terrestrial), expected limitations, robustness under noisy or low contrast recordings, and comparisons to existing open source alternatives at the level of usability and flexibility.
2. In the Materials and Methods section, I missed some relevant information. The manuscript gives light/dark cycles but does not report illumination spectra (except for IR) or intensities during recording. These details may affect tracking quality and should in my opinion be included in the main text. Likewise, for reproducibility, it would be helpful to specify the camera model, lens type, working distance / field of view, and characteristics of the IR illumination.
3. The Results section includes several statistical tests (t tests, correlations, etc.), but the Materials and Methods do not contain a dedicated statistics subsection. Adding one (including tests used, software packages, significance thresholds) would improve transparency.

Overall, I very much enjoyed reading this manuscript and commend the authors on their work! Please find some more specific comments below.

ABSTRACT

No comments - clear and concise.

INTRODUCTION

Page 2, line 33: "disrupts performance of this assay in zebrafish" - should this be reversed ("disrupts performance of zebrafish in this assay")?

RESULTS

No major comments - figures are clear and the text is logically structured.

DISCUSSION

As noted above, the biological interpretation is interesting but somewhat obscures the software validation narrative. I suggest trimming or relocating the biological interpretations.

MATERIALS AND METHODS

Page 9, line 5: Could you please provide more details about the light exposure? (spectral characteristics, intensities)

Page 9, line 14: Could you please specify the type of camera and lens, the working distance/field of view, and the characteristics of the infrared light source?

Page 9, line 39: Should this section include a brief description of the statistical analyses performed (tests used, software packages, correction methods, significance thresholds)?

Reviewer's Responses to Questions**Experimental quality**

Does each figure have the proper controls?

If 'No', please indicate reasons in Comments for Author box below.

Reviewer #1:

- Yes

Reviewer #2:

- Yes

Were the data analyzed using appropriate statistical tests?

If 'No', please indicate reasons in Comments for Author box below.

Reviewer #1:

- Yes

Reviewer #2:

- Yes

Reproducibility

Were experiments performed using adequate number of biological replicates?

If 'No', please indicate reasons in Comments for Author box below.

Reviewer #1:

- Yes

Reviewer #2:

- Yes

Does the methods section provide sufficient detail to permit reproducibility?

If 'No', please indicate reasons in Comments for Author box below.

Reviewer #1:

- No

Reviewer #2:

- No
-

Completeness

Are the manuscript's conclusions supported by the data?

If 'No', please indicate reasons in Comments for Author box below.

Reviewer #1:

- Yes

Reviewer #2:

- Yes
-

Scholarship

Do the authors cite and discuss the merits of data that would argue for and against their conclusion?

If 'No', please indicate reasons in Comments for Author box below.

Reviewer #1:

- Yes

Reviewer #2:

- Yes
-

Does the manuscript title & abstract accurately reflect the contents of the manuscript, without hyperbole?

If 'No', please indicate reasons in Comments for Author box below.

Reviewer #1:

- Yes

Reviewer #2:

- Yes
-

First revision

Author response to reviewers' comments

Reviewer 1: Overall, this is a well written paper describing a new open-source software tool for high throughput tracking of zebrafish. The tool developed looks to be an improvement over prior work from the lab that relied on the use of proprietary software and thus promises to be more widely accessible. The authors use the tool to analyze locomotor activity over an extended period of time, in response to 'dark flashes', and during the Y-maze. The data are largely clear and compelling that the tool is successfully working.

- We thank the reviewer for their positive comments.

Major points

1. It would strengthen the paper quite a bit to include additional analysis and discussion of the

comparison between StrIPETrack and the prior system developed by this group using LabVIEW (Joo et al, 2020). While it's clear that both systems yield similar findings with respect to analyzing behavior in two clutches (Fig. 3), it is also clear from the figures that there are some differences in the shapes and variation in the data. For example, with the dark flash response, the numerical values are much higher using the LabVIEW approach versus StrIPETrack, and the variation seems larger with StrIPETrack. Other subtle differences, for example in the activity on the second bout also look different. While I do not think this is an issue for use of StrIPETrack, it behooves additional analysis such as: how correlated are the findings from the two systems on a more granular basis (e.g., at the second by second timescale), and why are the absolute values so different for the dark flash response? I think this is an opportunity to discuss what is similar and different in terms of the algorithms used for tracking that may lead to these differences. This is important because it is rarely discussed in the literature how specific algorithms may bias the data in one direction or the other. Using the data the author's have generated would make the paper more impactful by providing a direct comparison and discussion of different approaches to tracking.

- Although the larvae assessed with the *StrIPETrack* and LabVIEW systems are from the same clutch, they are not the same animals and are tested in separate behavioral apparatuses. The largest observed difference, as pointed out by reviewer, is in the dark flash response. This data comes from post-hoc tracking of high-speed (285 fps) videos that are analyzed identically. Therefore, we cannot determine tracking differences between the StrIPETrack and LabVIEW software with the dark flash response data. We attribute the subtle phenotype differences not to systemic differences in the two software programs but instead to differences in the box environment and individual variability. For example, the placement of the IR light can affect contrast. We have added the following detail to the Figure 3 legend: ["Dark flash tracking is performed post-hoc and remains consistent between *StrIPETrack* and LabVIEW; numerical differences are likely explained by lighting differences between behavior tracking boxes."]. As the reviewer mentions below, controls should be matched to the experimental group as closely as possible to address any potential systemic differences.
- We agree that more carefully quantification of the same data from the two programs is an important comparison. We have added a new Supplementary Figure and the following outcome to Results section: "There was a strong, positive correlation between the dpix calculated by *StrIPETrack* and LabVIEW tracking ($r=0.973$, $p < 0.0001$). Consistency was moderate between the two tracking softwares ($ICC(3,1)=0.513$, $p < 0.0001$) (Fig. S1)."
- We have additionally compared our software to two existing programs (Stytra and BonZeb) and the following outcome to the Results section: "Although both BonZeb and Stytra are highly customizable and useful for multi-animal tracking, neither performed well with our below-described Y-maze dataset (Table S2). BonZeb performed better than Stytra but tracked many non-existent turns because the fish position was inaccurately switching arms."

2. The author's find that behavior can differ subtly from clutch to clutch. I think this is an important finding (that replicates their previous work) because it strongly suggests intra-clutch controls are essential. However, what is less clear to me is how much of the difference in behavior is driven by factors within the experimenters control. For example, in figure S3 the author's show that body size influences FMP Y-maze performance. I also see that the conditions for raising the fish are not well standardized (e.g, it's stated that fish are put into 150 mm dishes at a density of less than 160 fish per dish and then lower than 50/tank at 5dpf). Is it possible that fish in less dense dishes grew at a faster rate than those raised at higher density? Could this account or contribute to the clutch-to-clutch variability? I think this point needs to be discussed to better interpret this data.

- We thank the reviewer for their thoughtful comment on animal husbandry conditions. The question regarding how density impacts development was thoroughly investigated in a paper from our group (Joo et al., 2020). Density had no impact on behavior (160 fish/150mm dish vs 60-70). Further, in the experiment where we tested two clutches of fish, the fish were generated from two separate breeding pairs at the same time and treated equally. We also added the following information about animal husbandry: "Larvae at 5 dpf are fed live rotifers until 12 dpf. At 12 dpf, under slow water drip, hatched artemia and GEMMA 75 are fed twice per day until 60 dpf, and once per day on

weekends and holidays.”

3. A better measure to capture the consistency between manual and StriPETrack in Fig. 4 would be interrater reliability via the intraclass correlation coefficient (ICC). I suggest including this in addition to, or in lieu of, the Pearson's R.

- We have added an ICC2 analysis in addition to the Pearson's correlation: “There was a strong, significant positive correlation between the two methods ($r=0.903$, $p<0.0001$), and inter- method agreement was high ($ICC(2,1) = 0.899$, $p<0.0001$) (Fig. 4A).”

4a. The ReadMe file at <https://github.com/camden-cummings/behavior-analysis-scripts> leaves a bit to be desired and should be expanded a bit to make it easier for users to use that do not have a lot of Python experience to use.

- The additional information provided in Supplementary Methods has been added to the behavioral analysis README, as has an explanation of the different kinds of graphs made by the analysis, and the different variables within it. Additionally, we now include installation and use instructions, alongside a requirements.txt. This analysis software is described in more detail in the supplement of Joo et. al, 2020.

4b. While not necessary, I would recommend including a YouTube tutorial or something similar to post on the github page to help potentially new users familiarize themselves with this tool. Otherwise, the sparseness of the github page will likely be a barrier for those not as familiar with computers to engage with the tool.

- We thank the reviewer for the helpful comment. A video has been added to the GitHub (https://github.com/camden-cummings/roi_selector_dearpygui) to show the capabilities of the software. We have added the following statement to the Figure 1 legend: “For additional instruction, see the video in the GitHub repository.”

Minor points

5. How tedious is it for the user to draw all the different ROI's for either a 96 well plate or the Y-maze? It wasn't clear to me how much upfront time for the user would be needed to achieve what the authors have done.

- We have expanded the explanation of how to generate ROIs from the user perspective (Figure 1 legend): “For a 96-well plate, a user would first enter 11 into the ‘Vertical Line’ box & 7 in ‘Horizontal Line’. If any lines needed to be moved to match the cell walls, they could be shifted as a group with the WASD keys, or individually by with a drag and drop interface. Then, the user would hit ‘Generate ROIs’, making individual ROIs bounded by each line. These ROIs can be moved, deleted, or changed as in the typical interface.” In addition, a video tutorial is now available in the GitHub repository.

6a. How old were the fish used in Fig. 3? I did not see this in the methods

- We now include this detail in the Methods: “Larval zebrafish aged 4 dpf were placed in individual wells”.

6b. For the fish in Fig. 3, are these fish from different clutches from the same breeders? Or completely different breeders? If from different breeders, were they obtained on the same day and raised in parallel? More detail here would help better interpret these experiments.

- We now include this detail in the Results section: “Two separate breeding pairs were crossed to generate two clutches of fish, and fish from these clutches were placed in alternate columns of our 96-well behavior plates.”

6c. No strain or source of the fish is indicated in the methods.

- We now include this detail in the Methods: “All zebrafish were of the Ekkwill (EK)

background.”

6d. What are the ages and strains of fish in Fig. S2

- We now include this detail in the Fig. S2 legend: “21 dpf zebrafish”.

7a. What are the n's for Fig. S2, and what do the error ribbons represent?

- We have extended the Fig. S2 legend to include these details: “Variation in 21 dpf wild type zebrafish from different lines. Alternation tetragram percentage in 4 different lines of wild type zebrafish over 90 minutes of tracking. The mean alternation percentage for each 10 minute bin with the SEM is represented. (a)/red n=38. (b)/blue n=61, (c)/purple n=39, (d)/black n=46”.

7b. What type of camera was used for capturing the videos.

- We have added the following line to our Methods: “Videos were recorded using the FLIR FlyCapture SDK36 software on a Grasshopper GigE camera at 30 frames per second with H.264 compression using a streaming capture setting.”

7c. Line 27 of pg. 9, what does ABS stand for? Please spell it out if it is not going to be used anywhere else in the paper (I didn't see it anywhere else).

- We have modified this line to include the full name: “We print the Y-Mazes using white ABS (acrylonitrile butadiene styrene), as it is both resilient and cost-effective.”

8a. In the caption for Fig. 5 it says error bars represent SEM. But what about the error ribbons for parts B through E, and in E iii this looks like box and whisker plots which is usually something like median and interquartile range; the edge of the boxes and whiskers should be clearly defined.

- We have clarified the error bars in figure legend: “Error bars represent S.E.M. in A-C and Ei and Eii, and IQR in Eiii”

8b. In Figure 5, it's stated that one-sample t-tests were used to compare against random (12.5%), were these corrected for multiple comparisons? The text of the paper seems to suggest this (pg. 6, lines 3-5), but the figure caption does not.

- We have added a statistics section to summarize the statistics used in this paper for clarity, and added “and corrected for multiple comparisons using FDR” to the Figure 5 legend.

8c. Given the spread in the use of open-source machine learning approaches to tracking (e.g., DeepLabCut) in the past 7-8 years, a discussion in why this system is different than such popular systems would be useful for readers as they make a decision about what approach to take.

- We have added a paragraph in to the Discussion to address these comments about comparison to existing alternatives, including machine learning approaches: “Compared to other software, StrIPETrack was able to accurately track zebrafish in a variety of behavioral paradigms. Similar to our previous LabVIEW-based program (Joo et al., 2020), it also detected subtle differences in larval behavior. StrIPETrack was also able to quantify tetragrams in a complex juvenile navigation task, while neither BonZeb nor Stytra achieved comparable accuracy (Table S2). Because both Bonsai and Stytra are highly customizable, it is possible that a different set of parameters or a modification of the code used would improve the results; however, the default settings did not yield robust tracking. Other options are available, such as machine learning-based approaches (Barreiros et al., 2021; Mathis et al., 2018; Teicher et al., 2025), although they also have constraints. Machine learning typically requires substantial training and the creation of annotated datasets prior to tracking, and these methods can be too resource-intensive for real-time tracking. In contrast, StrIPETrack performs well with limited computational

resources and does not need to be trained. For quantifying more complex behaviors that cannot be fully captured by simple object tracking, such as posture dynamics, deep learning and semantic segmentation approaches may be more suitable.”

Reviewer 2: This manuscript presents StrIPETrack, a Python based, modular animal tracking software designed for flexible use in both simple and complex behavioral assays. The authors validate the software by comparing it against their previous LabVIEW-based pipeline in larval zebrafish assays and demonstrate its applicability to Y maze behavioral tasks. The topic is timely, the manuscript is very well written, and the software represents a valuable tool for behavioral biologists.

I review this work as a regular end user of animal tracking systems rather than a software development specialist (!), so my comments focus primarily on usability, clarity, methodological transparency, and how convincingly the validation demonstrates practical reliability.

Overall, in my opinion, this is a very strong contribution. My comments are mostly minor, with a few main suggestions for strengthening clarity and scope:

- We thank the reviewer for their positive comments.

1. While the biological findings (age related differences, size effects, etc.) are interesting, I feel that discussing these distracts from the primary purpose—validating the tracking software. I would find it more useful to expand the discussion of potential software performance in other species (fish vs. non fish, aquatic vs. terrestrial), expected limitations, robustness under noisy or low contrast recordings, and comparisons to existing open source alternatives at the level of usability and flexibility.

- An initial challenge we faced when designing our software was the prevalence of reflections and other low-quality video recordings in our dataset. Therefore, we optimized the software to be robust to low contrast and noisy videos. We have updated our hand-tracked data comparison to include an additional low contrast video, and demonstrated that it tracked correctly, as shown in the new Supplementary Fig. S3.
- Many tracking methods designed particularly for rodents perform additional analyses (e.g., complex interactions between animals). For our limited and less complex application, we optimized high modularity, high correctness, and speed. We expect that it is most useful for aquatic species and have added the following sentence to the discussion: “Additionally, it is easily extensible to most aquatic species, and its modularity would support other real-time or 3- dimensional tracking tasks (e.g., novel tank assay) with minimal modification.”
- We have added a more explicit comparison with two open-source alternatives and include that result in the new Table S2. We include the following new paragraph to the Discussion regarding comparisons to existing alternatives: “Compared to other software, StrIPETrack was able to accurately track zebrafish in a variety of behavioral paradigms. Similar to our previous LabVIEW-based program (Joo et al., 2020), it also detected subtle differences in larval behavior. StrIPETrack was also able to quantify tetragrams in a complex juvenile navigation task, while neither BonZeb nor Stytra achieved comparable accuracy (Table S2). Because both Bonsai and Stytra are highly customizable, it is possible that a different set of parameters or a modification of the code used would improve the results; however, the default settings did not yield robust tracking. Other options are available, such as machine learning-based approaches (Barreiros et al., 2021; Mathis et al., 2018; Teicher et al., 2025), although they also have constraints. Machine learning typically requires substantial training and the creation of annotated datasets prior to tracking, and these methods can be too resource-intensive for real-time tracking. In contrast, StrIPETrack performs well with limited computational resources and does not need to be trained. For quantifying more complex behaviors that cannot be fully captured by simple object tracking, such as posture dynamics, deep learning and semantic segmentation approaches may be more suitable.”

2. In the Materials and Methods section, I missed some relevant information. The manuscript gives light/dark cycles but does not report illumination spectra (except for IR) or intensities during

recording. These details may affect tracking quality and should in my opinion be included in the main text. Likewise, for reproducibility, it would be helpful to specify the camera model, lens type, working distance / field of view, and characteristics of the IR illumination.

- Catalog numbers are available in Joo et al., 2020, and we have added the following details to the Methods section: “A Grasshopper GigE camera with 50 mm fixed focal length lens with an IR filter was used for tracking, capturing fish illuminated with a long-range IR (850 nm) light. The light level from the white LED panel can be controlled as needed (Joo et al., 2020) (e.g., to induce a dark flash), and the baseline illumination level was approximately 410 lux in the larval behavioral assays. As previously (Cleal et al., 2021a), the Y-maze assays were run in the dark, which was shown to have minimal effect on navigation and yield consistent results.”

3. The Results section includes several statistical tests (t tests, correlations, etc.), but the Materials and Methods do not contain a dedicated statistics subsection. Adding one (including tests used, software packages, significance thresholds) would improve transparency.

- We thank the reviewer for their suggestion. We have now added a separate statistics section: “Statistical tests were conducted in Python. To compare *STRIPETrack*’s ability to calculate tetragrams, *STRIPETrack*-generated tetragrams and manually counted tetragrams were compared using a Pearson’s correlation, and agreement was assessed two-way random-effects intraclass correlation coefficient for absolute agreement (ICC2). To assess systemic bias between manual counting and *STRIPETrack*, a paired t-test was performed. Comparisons of alternation percent tetragrams to random chance (12.5%) were assessed using a one-sample t-test, corrected for multiple comparisons using FDR adjustments. A two-way ANOVA was used to compare the top 50% and bottom 50% of zebrafish by body size and, with a significant main effect of size, pairwise comparisons were calculated at each time point. Packages used for analysis include pingouin, seaborn, statsmodels.stats.multitest, scipy.stats, statsmodels.api and statsmodels.formula.api.”

Overall, I very much enjoyed reading this manuscript and commend the authors on their work! Please find some more specific comments below.

ABSTRACT

No comments - clear and concise.

INTRODUCTION

Page 2, line 33: “disrupts performance of this assay in zebrafish” - should this be reversed (“disrupts performance of zebrafish in this assay”)?

- This line has been changed as suggested.

RESULTS

No major comments - figures are clear and the text is logically structured.

DISCUSSION

As noted above, the biological interpretation is interesting but somewhat obscures the software validation narrative. I suggest trimming or relocating the biological interpretations.

- We thank the reviewer for this suggestion. Depending on the background and interest of the reader, some may find these interpretations valuable. For example, reviewer 1 was interested in how density impacts behavior and growth. Therefore, we chose to retain this discussion.

MATERIALS AND METHODS

Page 9, line 5: Could you please provide more details about the light exposure? (spectral characteristics, intensities)

- Catalog numbers are available in Joo et al., 2020, and we have added the following details to the Methods section: “A Grasshopper GigE camera with 50 mm fixed focal length lens with an IR filter was used for tracking, capturing fish illuminated with a long-range IR (850 nm) light. The light level from the white LED panel can be controlled as needed (Joo et al., 2020) (e.g., to induce a dark flash), and the baseline illumination level was approximately 410 lux in the larval behavioral assays. As previously (Cleal et al., 2021a), the Y-maze assays were run in the dark, which was shown to have minimal effect on navigation and yield consistent results.”

Page 9, line 14: Could you please specify the type of camera and lens, the working distance/field of view, and the characteristics of the infrared light source?

- Catalog numbers are available in Joo et al., 2020, and we have added the following details to the Methods section: “A Grasshopper GigE camera with 50 mm fixed focal length lens with an IR filter was used for tracking, capturing fish illuminated with a long-range IR (850 nm) light. The light level from the white LED panel can be controlled as needed (Joo et al., 2020) (e.g., to induce a dark flash), and the baseline illumination level was approximately 410 lux in the larval behavioral assays. As previously (Cleal et al., 2021a), the Y-maze assays were run in the dark, which was shown to have minimal effect on navigation and yield consistent results.”

Page 9, line 39: Should this section include a brief description of the statistical analyses performed (tests used, software packages, correction methods, significance thresholds)?

- We have added a separate statistical section.

Second decision letter

MS ID#: bio.062503R1

MS Title: StrIPETrack: a real-time, ROI-flexible tracking platform for high-throughput zebrafish behavior

Authors: Camden E. Cummings, Brandon L. Bastien, Jaqueline A. Martinez, Jinyue Luo and Summer B. Thyme

I am happy to tell you that your manuscript has been accepted for publication in Biology Open, pending our standard publication integrity checks. It was accepted on 30th March 2026.